# Use of Extracorporeal Life Support for Heart Transplantation: Key Factors to Improve Outcome

**DOI:** 10.3390/jcm10122542

**Published:** 2021-06-08

**Authors:** Jun Ho Lee, Nayeon Choi, Yun Jin Kim, Kiick Sung, Wook Sung Kim, Darae Kim, Jeong Hoon Yang, Eun-Seok Jeon, Sung Ho Shinn, Jin-Oh Choi, Yang Hyun Cho

**Affiliations:** 1Department of Thoracic and Cardiovascular Surgery, Hanyang University Seoul Hospital, Hanyang University College of Medicine, Seoul 04763, Korea; ecmo1984@gmail.com; 2Biostatistical Consulting and Research Lab, Medical Research Collaborating Center, Hanyang University, Seoul 04763, Korea; nayeon@hanyang.ac.kr (N.C.); yeun0148@hanyang.ac.kr (Y.J.K.); 3Department of Thoracic and Cardiovascular Surgery, Samsung Medical Center, Sungkyunkwan University School of Medicine, Seoul 06351, Korea; kiick.sung@samsung.com (K.S.); wooksung.kim@samsung.com (W.S.K.); 4Division of Cardiology, Department of Internal Medicine, Samsung Medical Center, Sungkyunkwan University School of Medicine, Seoul 06351, Korea; darae0918.kim@samsung.com (D.K.); jhysmc@gmail.com (J.H.Y.); eunseok.jeon@samsung.com (E.-S.J.); 5Department of Thoracic and Cardiovascular Surgery, Cheju Halla General Hospital, Jeju 63127, Korea; shinnsungho@gmail.com

**Keywords:** extracorporeal life support, bridge to transplantation, bridge to candidacy, heart transplantation, left ventricular assist device

## Abstract

Although patients receiving extracorporeal life support (ECLS) as a bridge to transplantation have demonstrated worse outcomes than those without ECLS, we investigated the key factors in the improvement of their posttransplant outcome. From December 2003 to December 2018, 257 adult patients who underwent heart transplantation (HTx) at our institution were included. We identified 100 patients (38.9%) who underwent HTx during ECLS (ECLS group). The primary outcome was 30-day mortality after HTx. The median duration of ECLS was 10.0 days. The 30-day mortality rate was 3.9% (9.2% in peripheral ECLS, 2.9% in central ECLS, and 1.9% in non-ECLS). The use of ECLS was not an independent predictor of 30-day and 1-year mortality (*p* = 0.248 and *p* = 0.882, respectively). Independent predictors of 30-day mortality were found to be higher ejection fraction (*p* < 0.001), Sequential Organ Failure Assessment score (*p* < 0.001), and total bilirubin level (*p* = 0.005). In a subgroup analysis, cannulation type was not a predictor of 30-day mortality (*p* = 0.275). Early ECLS application to prevent organ failure and sophisticated management of acute heart failure may be important steps in achieving favorable survival after HTx.

## 1. Introduction

Venoarterial (VA) extracorporeal life support (ECLS) is a type of temporary mechanical circulatory support (MCS) for patients in cardiogenic shock [1]. ECLS can be initiated for either newly diagnosed acute cardiac failure or decompensated chronic heart failure in patients already awaiting heart transplantation (HTx) [2]. Although a durable left ventricular assist device (LVAD) is a choice of MCS for bridge to transplantation (BTT), it has several contraindications, such as intolerance for a vitamin K antagonist, poor right ventricular function, restrictive cardiomyopathy, severe intracardiac problems, and patient refusal to use long-term MCS. Furthermore, the health care system, availability of devices, patient’s insurance policy, sociocultural background, and organ transplantation system vary widely by region and country [3].

Although the outcomes of HTx performed directly after ECLS have been poor [4,5], careful patient management in an experienced center showed a favorable outcome [6]. Because the highest priority of HTx in ECLS has been established in most countries, the short waiting time may justify direct HTx from ECLS [7,8,9]. However, the factors that improve post-HTx survival with the liberal use of ECLS as a BTT are not well known. Hence, we reviewed our 15-year experience of HTx to determine the key factors that improve outcome.

## 2. Methods

### 2.1. Study Patients

A total of 262 adult patients underwent HTx at Samsung Medical Center from December 2003 to December 2018 (Figure 1). All patients were on the waiting list for HTx in the Korean Network for Organ Sharing (KONOS). We excluded patients who underwent repeated transplantation and those younger than 18 years. In addition, we excluded five patients who had undergone implantation of a durable LVAD before HTx. In the final cohort of 257 cases, 100 patients (38.9%), who were supported by VA ECLS or temporary LVAD while awaiting HTx, were assigned to the ECLS group. In 65 patients, peripheral cannulation was maintained until HTx, and these patients were assigned to the peripheral ECLS group. The other 35 patients who had central ECLS at the time of HTx were assigned to the central ECLS group.

### 2.2. Indication for ECLS Installation and Criteria for ECLS as a BTT

In patients who were already on the waiting list for HTx, the degree of organ failure, incidence of ventricular arrhythmia, and symptoms of low cardiac output and pulmonary edema were principally used in the decision to commence ECLS. In cases of deterioration of organ failure or symptoms related to heart failure, patients were closed monitored by both heart failure physicians and cardiac intensivists. The decision to commence ECLS was taken using a multidisciplinary team approach. Patients who could not be weaned from VA ECLS and met all following criteria were considered potential candidates of ECLS as a BTT: normal mentation; no irreversible organ failure; age < 70 years; absence of active infection; absence of severe pulmonary hypertension; no recent history of malignancy; and good social support. Final listing for HTx was made after discussion among the multidisciplinary team.

### 2.3. Management and Cannulation Strategies of ECLS

Durable LVAD was not covered by Korean National Insurance before October 2018. Therefore, most patients who failed to respond to medical therapy or intra-aortic balloon pump underwent ECLS as BTT. Patients at our institution with acute and chronic heart failure were observed by a multidisciplinary heart failure team, which was formally established in 2014. Since 2014, all patients have been provided care under the updated guidelines of modern critical care, including prevention and management of pain, agitation, delirium, immobilization, and sleep deprivation [10].

The establishment of peripheral VA ECLS using Seldinger’s technique at our institution has been described previously [1]. When the left ventricle was distended, atrial septal puncture was performed by interventionists in a catheterization laboratory (*n* = 17). We have not used intra-aortic balloon pumps as a means of left heart decompression [11,12]. In 18 patients (51.4%) in the central ECLS group, peripheral VA ECLS was initially implemented and later converted to central ECLS. Another 17 patients (48.6%) in the central ECLS group underwent central cannulation from the beginning of ECLS. In our institution, surgical left heart decompressive procedures are typically performed during central cannulation. The strategy of left heart decompression at our institution has been previously published [13]. Among these 35 patients in the central ECLS group, 10 patients (28.6%) had LVAD-type cannulation using an ECLS device. We avoided mechanical ventilation and immobilization as much as possible in accordance with the stability of the cannulation site and the patients’ general condition. This strategy enables patients who are receiving ECLS to wait for HTx with a minimal risk of ventilator-associated pneumonia and complications related to peripheral cannulation.

### 2.4. Endpoints and Follow-Up

The primary outcome of the study was 30-day mortality. All patients were monitored after surgery by HTx physicians at Samsung Medical Center. Baseline characteristics of clinical data were collected from medical records and databases. We acquired follow-up clinical data, including vital status, through a review of medical records and telephone interviews. To complete the data, including mortality, we confirmed information by the National Registry of Births and Deaths using the unique personal identification number for each patient.

### 2.5. Statistical Analysis

Baseline characteristics, anthropometric, and clinical characteristics of the study population are presented as either mean ± standard deviation, median with interquartile range (IQR), or frequency and proportion. Wilcoxon rank sum tests were used to compare skewed continuous variables. For categorical variables, the chi-square and Fisher’s exact tests were used to compare variables among the groups. The Cox proportional hazards regression model for univariable and multivariable analyses were used to determine independent predictors of 30-day mortality. Multivariable analyses were performed using a stepwise variable selection method, in which all variables with a *p* value of less than 0.15 were included in the univariable analyses. To estimate the survival curves during the follow-up period, we used the Kaplan–Meier method, and survival rates were compared among groups using the log rank test.

We adjusted for differences in baseline characteristics using weighted Cox proportional hazards regression models with inverse probability of treatment weighting (IPTW) to reduce potential confounding factors [14]. Variables for adjustment are summarized in Table A1. The standardized difference was calculated from the mean and prevalence for continuous and dichotomous variables, respectively, and the results are shown in Table 1.

All tests were two-tailed. *p* values of less than 0.05 were used to denote statistical significance. Statistical analysis was performed with the R programming language, version 3.5.1 (R Foundation for Statistical Computing, Vienna, Austria) and SAS version 9.4 (SAS Institute Inc., Cary, NC, USA).

### 2.6. Ethics

The study was conducted according to the guidelines of the Declaration of Helsinki, and approved by the Institutional Review Board of Samsung Medical Center (IRB No. SMC 2019-09-109). Informed consent from the study participants was waived due to the retrospective nature of this study.

## 3. Results

### 3.1. Baseline Characteristics

The median age of all patients was 53 years (range, 18–78 years), and 79 patients (30.7%) were female. The median duration of ECLS before HTx was 10.0 days (IQR, 7–17 days). In the ECLS group, 68 ECLS patients were on the waiting list for HTx at the time of VA ECLS implantation. In the other 13 ECLS patients, decompensated chronic heart failure was diagnosed after hospitalization. Another 19 ECLS patients were diagnosed with irreversible acute heart failure (Figure 1). Table 1 summarizes the baseline characteristics of the patients in the ECLS and non-ECLS groups and we found significant differences in several characteristics between the two groups.

### 3.2. Perioperative Outcomes and Predictors of 30-Day Mortality

Although cardiopulmonary bypass time and total ischemic time were similar in the two groups, postoperative complications were more frequent in the ECLS group than in the non-ECLS group. Table 2 summarizes the operative and postoperative data.

The 30-day mortality rate in the overall cohort was 3.9% (*n* = 10). After univariable adjustment including IPTW, 30-day mortality rates did not differ significantly between the two groups (7.0% in ECLS and 1.9% in non-ECLS; *p* = 0.248). The 1-year mortality rate in the overall cohort was 12.8% (*n* = 33). After univariable adjustment including IPTW, 1-year mortality rates did not differ significantly between the two groups (16.0% in ECLS and 10.8% in non-ECLS; *p* = 0.882). After multivariable adjustment including IPTW, Kaplan–Meier curve did not show a significant difference in mortality up to one year after HTx (Figure 2; log rank *p* = 0.991).

Table 3 summarizes the results of the Cox proportional hazards regression model for univariable and multivariable analyses of 30-day mortality after IPTW. Univariable analysis including IPTW indicated that the use of ECLS was not a predictor of 30-day mortality (*p* = 0.248; hazard ratio (HR) 2.132; 95% confidence interval (CI) 0.590–7.704). Multivariable analysis including IPTW indicated that left ventricular ejection fraction (LVEF; *p* < 0.001; HR 1.064; 95% CI 1.025–1.103), Sequential Organ Failure Assessment (SOFA) score (*p* < 0.001; HR 1.349; 95% CI 1.184–1.538), and the total bilirubin level (*p* = 0.005; HR 1.055; 95% CI 1.016−1.095) were independent predictors of 30-day mortality.

### 3.3. Subgroup Analysis of the ECLS Group According to Cannulation Type

Table 4 summarizes the baseline characteristics and postoperative data of patients in the peripheral and central ECLS groups. The durations of ECLS and ventilator support before HTx were longer in the central ECLS group compared to the peripheral ECLS group.

No significant differences in 30-day outcomes were observed between the two groups (9.2% mortality in the peripheral ECLS group and 2.9% mortality in the central ECLS group; *p* = 0.275). Further, one-year outcomes did not differ significantly between the two groups (18.5% mortality in the peripheral ECLS group and 11.4% mortality in the central ECLS group; *p* = 0.379).

Kaplan–Meier analysis demonstrated no significant differences in mortality up to one year after HTx among the peripheral ECLS, central ECLS, and non-ECLS groups (Figure A1; log rank *p* = 0.179).

## 4. Discussion

Although the best option for BTT is durable LVAD, there are a few circumstances in which ECLS can be favored. First, in some countries, a durable LVAD is still not available or is too restrictive [15]. For example, the approval process by the National Health Insurance in Korea takes a few weeks. Second, when a patient has severe right ventricular dysfunction or restrictive cardiomyopathy, the implantation of durable LVAD is not a viable option [16,17]. Furthermore, in most national organ-sharing systems, highest priority for HTx is given to patients on temporary MCS such as ECLS. In such transplantation systems, the waiting time on ECLS may be within the safe range, namely, less than a few weeks, if a high level of ICU and ECLS care is provided.

Although post-HTx complications were more common in the ECLS group than in the non-ECLS group, there was no statistically significant difference in 30-day mortality between the two groups even after adjustment by IPTW. High LVEF, high SOFA score, and high total bilirubin level were found to be independent risk factors. Thus, timing of ECLS insertion is important, and ECLS should be deployed before severe multiorgan failure, including hepatic failure. The finding that high LVEF was an independent predictor of 30-day mortality was interesting. We believe that patients with relatively high LVEF have developed acute heart failure due to acute myocardial infarction, myocarditis, hypertrophic cardiomyopathy at burn-out stage, or restrictive cardiomyopathy. These conditions, diseases, and sequelae may not be favorable to HTx. Therefore, we suggest special attention be paid to patients with irreversible acute heart failure.

The most important step to improve outcomes of HTx is timely ECLS initiation. The second step would be applying sophisticated ECLS and up-to-date intensive care to prevent both ECLS and intensive care-related complications. For patients who are waiting for HTx while on ECLS, the final step could be the liberal use of central cannulation. In some relatively stable patients who require a prolonged waiting time for HTx, primary central ECLS can be performed. In other patients who already had peripheral ECLS, cannulation can be stitched to a central type after careful discussion and hemodynamic stabilization. Central cannulation and circuit configurations vary according to the patient’s right ventricular and pulmonary function. In general, we consider central conversion on the 14th day of peripheral cannulation. Figure 3 shows our recent strategy.

Moonsamy et al. also reported that temporary circulatory support (TCS)-VAD had a survival advantage over ECLS and was similar to durable LVAD as a BTT in the United Network of Organ Sharing (UNOS) database [7]. In their study, there is no information about cannulation type or ECLS duration. In our study, the definition of ECLS includes TCS-VAD described in Moonsamy’s study, because 35.0% of our ECLS patients had nonperipheral cannulation, including central VA ECLS and temporary VAD with/without membrane oxygenator TCS-VAD. In other words, in an emergency setting, temporary VAD implantation cannot be placed as conveniently and quickly as ECLS. We believe that some patients in the TCS-VAD group probably had been converted from peripheral ECLS, like our patients. Therefore, we believe that various cannulation options should be offered to patients, including either direct central cannulation or peripheral ECLS first. To reduce peripheral cannulation-related complications, the timely conversion to central cannulation is critical.

Coutance et al. also compared ECLS-bridged HTx and non-ECLS-bridged HTx in their large retrospective study [6]. Although they emphasized strong ECLS care, including the prevention of ECLS complications and the application of light sedation, their selection criteria for HTx in ECLS patients seems to be quite strict. In contrast to our procedure, they performed HTx only for patients without other organ failure. Because the study by Coutance et al. included only relatively low-risk patients [6], it is difficult to determine how we can improve the outcome of HTx. Table 5 presents a comparison of previous studies (Coutance et al. and Moonsamy et al.) with the current study [6,7]. Each paper showed various HTx waiting periods, and to improve HTx outcomes, it will be important to implement the optimal settings for BTT based on the circumstances of each country.

We used an approach to prevent the occurrence of ECLS-related complications. Most patients had a preventive distal limb perfusion catheter. For example, in the ECLS group, the rate of limb ischemia was 5.0% (Table 2). During the same period as our study, the rate of limb ischemia in all patients with VA ECLS at our institution was 5.69%. Taking into consideration that the rate of limb ischemia has been reported to be as high as 70% [18], our outcome is good. Intensivists have maintained the most up-to-date ICU care, such as light sedation, minimally invasive ventilation, and aggressive mobilization. High-quality ECLS and ICU care were also emphasized in the article by Coutance et al. [6]. Although post-HTx complications were more common in the ECLS group than in the non-ECLS group, they did not affect post-HTx survival.

### Study Limitations

As this study was retrospective and involved 257 patients at a single tertiary center, its statistical power may be limited. The duration of ECLS before HTx in the ECLS group was short (median: 10 days, mean: 16.0 ± 18.9 days). However, the duration was longer than in previous studies [6,19]. We did not include patients who were listed for HTx but did not receive transplantation. Thus, the study may have some selection bias.

Further, data regarding transfusion rates were not included in our database. Accordingly, patients in the ECLS group may have received greater blood transfusion volumes than those in the non-ECLS group, and some perioperative complications are known to be associated with rate of blood transfusion.

As implantable LVADs were not covered by Korean National Insurance until September 2018, the present study population did not have access to a LVAD [20]. In the era of implantable LVAD, ECLS is still used as a BTT strategy in South Korea, many European countries, many Asian countries, and North America. The number of patients using ECLS as BTT can vary according to national transplant systems and insurance policies. We believe that our clinical experience may be of value to many clinicians.

## 5. Conclusions

After adjustment including IPTW, the use of ECLS before HTx was not a risk factor of 30-day mortality. Degree of organ failure, particularly hepatic failure and acute heart failure, were independent predictors of 30-day mortality. Early ECLS application to prevent organ failure and sophisticated management of acute heart failure may be important steps in achieving favorable survival after HTx.

## Figures and Tables

**Figure 1 jcm-10-02542-f001:**
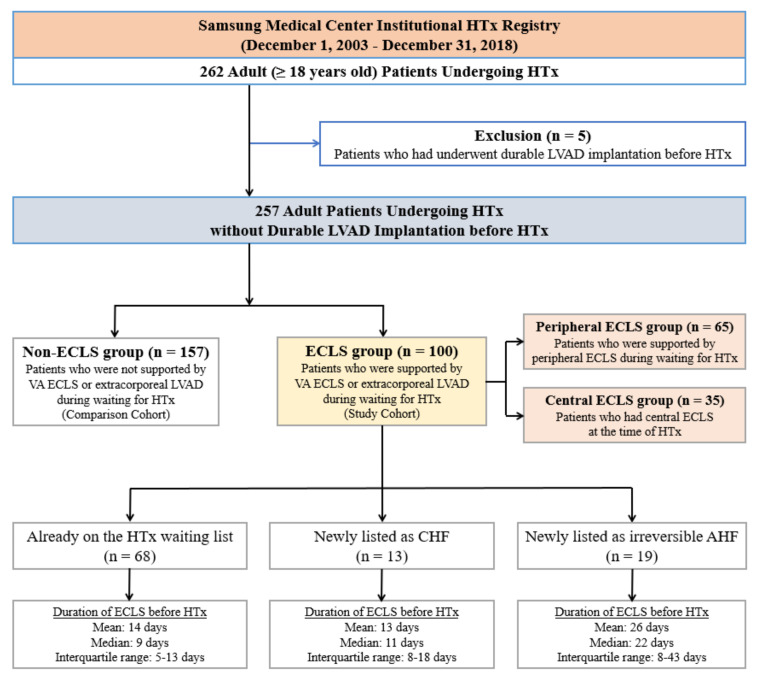
Flow diagram of patient recruitment. HTx, heart transplantation; LVAD, left ventricular assist device; ECLS, extracorporeal life support; VA, venoarterial; CHF, chronic heart failure; AHF, acute heart failure.

**Figure 2 jcm-10-02542-f002:**
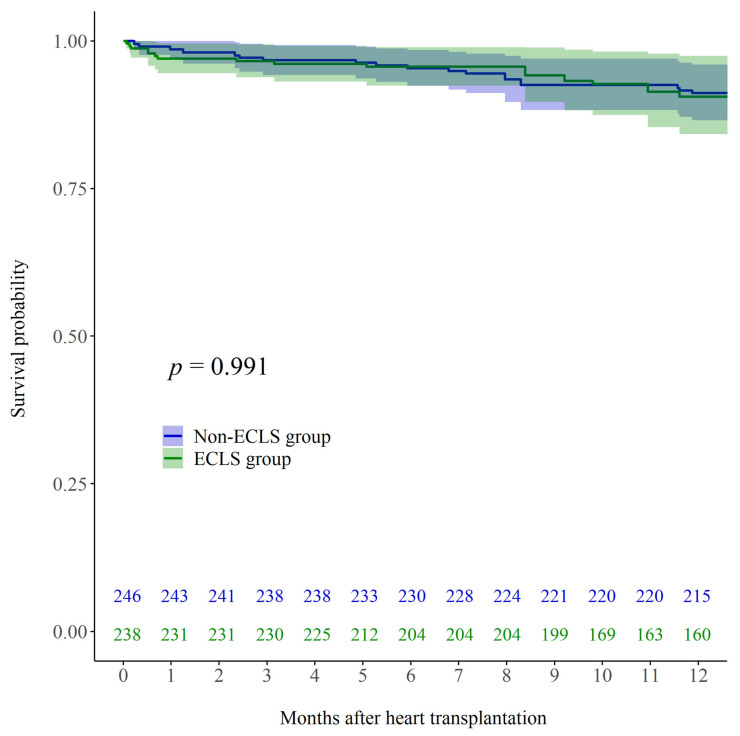
Kaplan–Meier post-HTx survival curves for patients who received ECLS while waiting for HTx (ECLS; green line) and those who did not receive ECLS before HTx (Non-ECLS; blue line) after multivariable adjustment including IPTW. HTx, heart transplantation; ECLS, extracorporeal life support; IPTW, inverse probability of treatment weighting.

**Figure 3 jcm-10-02542-f003:**
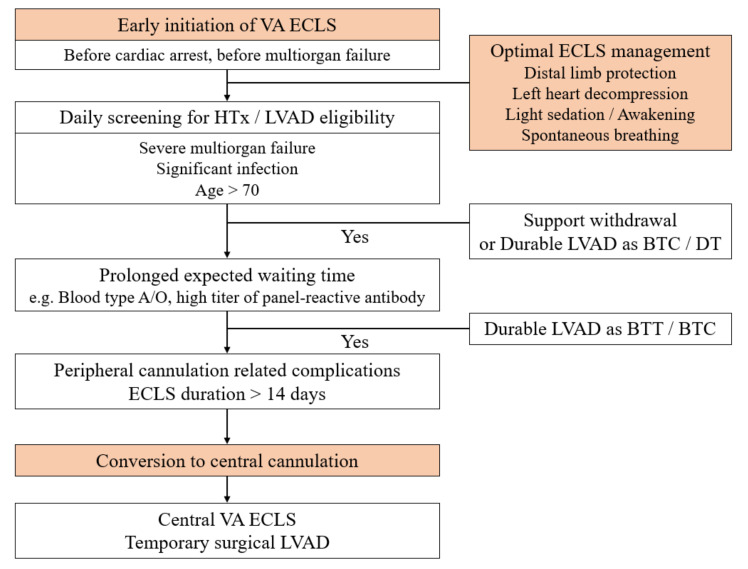
The recent strategy of ECLS as BTT in our institution. ECLS, extracorporeal life support; BTT, bridge to transplantation; VA, venoarterial; HTx, heart transplantation; LVAD, left ventricular assist device; BTC, bridge to candidacy; DT, destination therapy.

**Table 1 jcm-10-02542-t001:** Baseline characteristics of all patients.

	Overall Cohort				IPTW	
	ECLS Group (*n* = 100)	Non-ECLS Group (*n* = 157)	*p*-Value	Standardized Difference	*p*-Value	Standardized Difference
Age of recipient (years)	51.5 (38.5−59.5)	54 (44−61)	0.186	−0.183	0.875	0.015
Female recipients	29 (29.0)	50 (31.9)	0.630	−0.062	0.778	−0.026
Age of donor (years)	42.5 (33−50)	42 (34−49)	0.761	0.038	0.320	0.108
Female donors	22 (22.0)	46 (29.3)	0.196	−0.168	0.019	0.215
Gender mismatch	35 (35.0)	52 (33.1)	0.756	0.040	0.002	0.292
Diabetes	29 (29.0)	29 (18.5)	0.049	0.249	0.540	−0.056
Hypertension	27 (27.0)	52 (33.1)	0.300	−0.134	0.006	−0.252
Body mass index (kg/m^2^)	21.9 (19.2−24.2)	22.1 (20.4−24.5)	0.142	−0.195	0.643	−0.053
Previous cardiac surgery	23 (23.0)	17 (10.8)	0.009	0.329	0.782	−0.025
Previous PCI or CABG	28 (28.0)	32 (20.4)	0.159	0.179	0.003	0.270
Stroke	8 (8.0)	11 (7.0)	0.767	0.038	0.723	−0.032
Chronic renal insufficiency	9 (9.0)	33 (21.0)	0.011	−0.341	0.005	−0.257
Dialysis	30 (30.0)	13 (8.3)	<0.001	0.574	0.193	−0.119
CPR	31 (31.0)	11 (7.0)	<0.001	0.642	0.573	0.051
LVEF (%)	20 (16−25)	23 (19−30)	0.002	−0.303	0.826	0.036
DCMP	54 (54.0)	100 (63.7)	0.122	−0.198	0.061	−0.171
ICMP	26 (26.0)	29 (18.5)	0.151	0.182	0.026	0.204
SOFA score	7 (6−10)	3 (2−5)	<0.001	1.515	0.966	0.094
Total bilirubin (mg/dL)	2.6 (1.5−7.4)	1.3 (0.7−2.2)	<0.001	0.616	0.949	0.119
Pre-HTx hospital day	25 (12−43)	50.5 (23−92)	<0.001	-	-	-
Pre-HTx ECLS day	10 (7−17)	NA	NA	-	-	-
Pre-HTx ventilator day	7 (2−13)	0 (0−0)	<0.001	-	-	-
WBC (×10^3^/mm^3^)	10.6 (8.2−13.8)	6.5 (5.2−8.1)	<0.001	-	-	-
Hemoglobin (g/dL)	9.6 (8.8−10.4)	11.9 (10.5−13.1)	<0.001	-	-	-
Hematocrit (%)	28.7 (26.3−31.6)	36.3 (31.1−39.3)	<0.001	-	-	-
Platelet count (×10^3^/mm^3^)	97.5 (72−126)	181 (137−233)	<0.001	-	-	-
Creatinine (mg/dL)	1.1 (0.7−1.7)	1.1 (0.9−1.4)	0.722	-	-	-
Total protein (g/dL)	5.4 (4.9−5.8)	6.6 (6.2−7.1)	<0.001	-	-	-
Albumin (g/dL)	3.1 (2.8−3.3)	3.9 (3.6−4.2)	<0.001	-	-	-
NT-proBNP (pg/mL)	9836 (3588−23,658)	4052 (2,263.5−8818)	<0.001	-	-	-

Non-normally distributed numerical variables are presented as medians (interquartile ranges) and were tested using the Wilcoxon rank-sum test. Categorical variables are presented as numbers (percentages) and were tested using the chi-square test or Fisher’s exact test. IPTW, inverse probability of treatment weighting; ECLS, extracorporeal life support; PCI, percutaneous coronary intervention; CABG, coronary artery bypass graft surgery; CPR, cardiopulmonary resuscitation; LVEF, left ventricular ejection fraction; DCMP, dilated cardiomyopathy; ICMP, ischemic cardiomyopathy; SOFA, Sequential Organ Failure Assessment; HTx, heart transplantation; NA, not applicable; WBC, white blood cell; NT-proBNP, N-terminal prohormone of brain natriuretic peptide.

**Table 2 jcm-10-02542-t002:** Operative and postoperative characteristics of all patients.

	ECLS Group (*n* = 100)	Non-ECLS Group (*n* = 157)	*p*-Value
CPB time (min)	143 (123.5−170)	146 (127−182.5)	0.136
ACC time (min)	78.5 (64−96.5)	88 (70−106)	0.005
Total ischemic time (min)	179.5 (146.5−237)	180 (147.5−234.5)	0.901
Cold ischemic time (min)	119 (85.5−179.5)	114 (81−162.5)	0.399
Warm ischemic time (min)	59 (49.5−70)	66 (56−79)	<0.001
Postoperative ICU stay (days)	16 (11−28)	9 (6−13)	<0.001
Total hospital stay (days) *	75 (54−125)	79 (52−128)	0.935
Ventilator support ≥ 3 days after HTx	37 (37.0)	14 (8.9)	<0.001
Reoperation for bleeding control	17 (17.0)	11 (7.0)	0.012
Infection	36 (36.0)	17 (10.8)	<0.001
Limb ischemia	5 (5.0)	NA	NA
30-day mortality	7 (7.0)	3 (1.9)	0.248
1-year mortality	16 (16.0)	17 (10.8)	0.882

* Total hospital stay includes pre- and post-HTx hospital stay. Non-normally distributed numerical variables are presented as medians (interquartile ranges) and were tested using the Wilcoxon rank-sum test. Categorical variables are presented as numbers (percentages) and were tested using the chi-square test. *p* values of 30-day and 1-year mortality were determined by the Cox proportional hazards regression after adjustment including inverse probability of treatment weighting. ECLS, extracorporeal life support; CPB, cardiopulmonary bypass; ACC, aorta cross-clamping; ICU, intensive care unit; HTx, heart transplantation; NA, not applicable.

**Table 3 jcm-10-02542-t003:** Independent predictors of 30-day mortality after IPTW.

	Univariable				Multivariable			
	*p*-Value	Hazard Ratio	95% Confidence Interval		*p*-Value	Hazard Ratio	95% Confidence Interval	
			Lower 0.95	Upper 0.95			Lower 0.95	Upper 0.95
Pre-HTx ECLS	0.248	2.132	0.590	7.704				
Age of recipient (years)	0.368	1.024	0.973	1.077				
Female recipients	0.258	2.008	0.601	6.716				
Age of donor (years)	0.664	1.013	0.957	1.072				
Female donors	0.188	0.363	0.080	1.642				
Gender mismatch	0.290	1.918	0.574	6.414				
Diabetes	0.362	1.827	0.500	6.680				
Hypertension	0.993	0.000	0.000	NA				
Body mass index (kg/m^2^)	0.344	1.069	0.931	1.228				
Previous cardiac surgery	0.012	4.864	1.408	16.805				
Previous PCI or CABG	0.571	0.655	0.152	2.830				
Stroke	0.993	0.000	0.000	NA				
Chronic renal insufficiency	0.353	1.990	0.466	8.497				
Dialysis	0.036	3.722	1.089	12.716				
CPR	<0.001	10.328	2.996	35.602				
LVEF (%)	0.008	1.049	1.012	1.086	<0.001	1.064	1.025	1.103
DCMP	0.053	0.266	0.070	1.014				
ICMP	0.290	0.369	0.058	2.338				
SOFA score	<0.001	1.411	1.256	1.585	<0.001	1.349	1.184	1.538
Total bilirubin (mg/dL)	<0.001	1.104	1.071	1.138	0.005	1.055	1.016	1.095

IPTW, inverse probability of treatment weighting; HTx, heart transplantation; ECLS, extracorporeal life support; PCI, percutaneous coronary intervention; CABG, coronary artery bypass graft surgery; NA, not applicable; CPR, cardiopulmonary resuscitation; LVEF, left ventricular ejection fraction; DCMP, dilated cardiomyopathy; ICMP, ischemic cardiomyopathy; SOFA, Sequential Organ Failure Assessment.

**Table 4 jcm-10-02542-t004:** Subgroup analysis of the ECLS group according to the cannulation type.

	Overall Cohort		
	Peripheral ECLS Group (*n* = 65)	Central ECLS Group (*n* = 35)	*p*-Value
Age of recipient (years)	50 (40−59)	52 (37−60)	0.911
Age of donor (years)	41 (33−50)	45 (35−51)	0.439
Dialysis	16 (24.6)	14 (40.0)	0.109
CPR	22 (33.9)	9 (25.7)	0.402
LVEF (%)	20 (15−28)	20 (17−23)	0.494
SOFA score	7 (6−10)	7 (6−9)	0.614
Total bilirubin (mg/dL)	2.4 (1.4−6.6)	2.7 (1.9−7.7)	0.232
Pre-HTx ECLS day	9 (6−13)	15 (8−25)	0.003
Pre-HTx ventilator day	5 (1−10)	9 (3−15)	0.014
30-day mortality	6 (9.2)	1 (2.9)	0.275
1-year mortality	12 (18.5)	4 (11.4)	0.379

Non-normally distributed numerical variables are presented as medians (interquartile ranges) and were tested using the Wilcoxon rank-sum test. Categorical variables are presented as numbers (percentages) and were tested using the chi-square test. *p*-values of 30-day and 1-year mortality were determined by the Cox proportional hazards regression. ECLS, extracorporeal life support; CPR, cardiopulmonary resuscitation; LVEF, left ventricular ejection fraction; SOFA, Sequential Organ Failure Assessment; HTx, heart transplantation.

**Table 5 jcm-10-02542-t005:** Comparison of recent and current studies.

	The Current Study (*n* = 100)	Coutance et al. [6] (*n* = 118)	Moonsamy et al. [7] (*n* = 177)
Study setting	Single institution Retrospective	Single institution Retrospective	Multi-center registry Retrospective
Study period (duration)	2003−2018 (15 years)	2012−2016 (5 years)	2005−2017 (13 years)
Age of recipient (years)	48.8 ± 14.2	48.0 ± 12.4	46 ± 15
Age of donor (years)	40.9 ± 11.7	44.9 ± 15.3	32 ± 12
Previous cardiac surgery	23 (23.0)	23 (19.5)	82 (46.3)
Dialysis	30 (30.0)	1 (0.9)	32 (18.1)
ICMP	26 (26.0)	37 (31.3)	40 (22.6)
Mechanical ventilation	84 (84.0)	13 (11.0)	82 (46.3)
Total days on the waiting list	48.9 ± 94.0	18.0 ± 56.0	89 ± 214
Pre-HTx ECLS days	10 (7−17)	9 (5−15)	NA
Creatinine (mg/dL)	1.5 ± 1.6	0.8 ± 0.4	1.3 ± 0.8
Total bilirubin (mg/dL)	6.3 ± 9.7	5.6 ± 4.1	2.3 ± 3.4
Total ischemic time (minutes)	192.9 ± 62.3	199 ± 47	198 ± 60
1-year survival of ECLS group (%)	84.0	85.5	67.6
1-year survival of no support (%)	89.2	80.7	90.2

Numerical variables are presented as medians (interquartile ranges) or means ± standard deviations. Categorical variables are presented as numbers (percentages). ICMP, ischemic cardiomyopathy; HTx, heart transplantation; ECLS, extracorporeal life support; NA, not applicable.

## Data Availability

The authors confirm that the data supporting the findings of this study are available within the article and its supplementary materials. Raw data that support the findings of this study are available from the corresponding author, upon reasonable request.

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
