# Peer review of "Use of Extracorporeal Life Support for Heart Transplantation: Key Factors to Improve Outcome"

_jcm, 2021, doi:10.3390/jcm10122542_

Round 1

Reviewer 1 Report

The study from Lee and coworkers reports among a consecutive series of 257 adult HTx recipient over a 15-years period, analyzing the impact of ECLS as a direct bridge to HTx. The topic is extremely interesting especially in the acute HF setting (post acute MI cardiogenic shock, fulminant myocarditis). In this scenario, ECMO team members are often in need to acutely support a largely unknown patient and then, if recovery of cardiac function does not occur, they should proceed towards a long-term MCS and/or HTx trying to establish the role of multiorgan failure and/or uncertain neurological status.

The authors are to be congratulated for their series and for their results. However, I think that there are some major issues to be addressed in the manuscript before considering for publication.

  • I do not think that 1-year mortality is the right primary study outcome, or, at least, is not the most important one. I personally think that the most important missing information in the manuscript is the in-hospital mortality, or the 30-days and 90-days mortality. By looking to the preoperative characteristics of the study population, if we do not comment on the perioperative outcomes, any other consideration on the follow-up is misleading. I’ll explain in the following comments.
  • It is clear that preoperative profile of the two study groups (ECLS vs non-ECLS) is completely different, and this is consistent to the clinical practice. In particular, they present with more multiorgan failure (MOFS), they have more anemia (9.6 vs 11.9) and so on. However, the description of CKD is misleading. The authors reported that in ECLS group there is less CKD rate (9% vs 30%) but indeed 21% are in chronic dialysis (vs 8%).
  • The perioperative outcomes are poorly described. It is evident that ECLS group has worse outcomes (Pulmonary dysfunction rate= 37% vs 9%, Bleeding rate=17% vs 7%, Infection rate= 36% vs 11%). This data are completely expected, but perioperative mortality rate is completely missing. If I look to the KM survival curve, there is a significant step at the beginning of the time scale, which I guess reflects a significant difference in hospital mortality. Furthermore, which is the transfusion rate in the two groups? I guess that ECLS group undergo much more blood transfusion and, indeed, many perioperative complications seem to be transfusions-related.
  • In this viewpoint, discussion should be completely revised. Patients supported with ECLS as a direct bridge to HTx are a very high-risk population for several reasons. First, they are often acute patients, which are often supported in the setting of cardiac arrest. Second, they often miss a careful evaluation for HTx eligibility before support, due to emergent clinical presentation, and they often present with uncertain neurological status during ECLS. The decision to proceed to advanced form of MCS is difficult in this setting. In western countries scarce donor availability often preclude the possibility of direct bridging from ECLS to HTx, especially for the fears to “waste” a suitable organ in uncertain conditions. I understand that in the authors country LVAD are not easily available from an economic viewpoint, but this reason cannot force a clinical indication and this decision-making process cannot be extended in foreign countries where ECLS bridge to LVAD could be a more reasonable option.
  • Indications favoring ECLS vs LVAD in acute HF setting are generally shareable, but I partially disagree with the authors regarding biventricular or RV failure. These conditions are contraindications to LVAD, but, if RHF is related to pulmonary hypertension, it can be a contraindication for HTx too. In my viewpoint, in case of acute HF and Cardiac arrest, peripheral ECLS is the fast, cheap and basic solution for MCS, in order to re-establish circulation, reverse organ failure and assess neurological status. If these conditions are achieved, and no cardiac recovery occurs, patient can be bridged to more advanced form of MCS or HTx.
  • Another relevant point regards the ECLS setting. I personally believe that in acute HF, ECLS should be implanted peripherally as much as possible. Central cannulation carries higher rate of bleeding and infection, makes a second sternotomy (for LVAD or HTx) more troublesome and prevent patient mobilization during ECLS support. Which were the reasons for converting from peripheral to central? Which was the rate of complications between the two settings, and especially of limb complications? Do the authors have experience of subclavian cannulation?
  • Finally, the authors in ECLS group included 35 central and 65 peripheral ECLS. I think that they should make a three group comparison (peripheral ECLS, vs central ECLS, vs no ECLS) and look for perioperative outcomes. In my personal experience, peripheral ECLS have less complications, in term of bleeding, transfusion and infection rate. So it could be possible that in this group, perioperative complications after HTx are less than central ECLS group and comparable with no-ECLS group.

Reviewer 2 Report

The paper by Lee at al. reports the Author’s single-centre retrospective experience on ECLS as a bridge to transplant and investigates its impact on outcome. The Authors retrospectively investigate a 15-year period (2003-2018) on 257 patients, 100 of whom required a ECLS before transplant. Opposite to what is commonly known and contrary to well established literature data, they report that ECLS did not impact 1-year mortality. Furthermore, they report an outstanding 1-year mortality of only 12.8% in a very high risk category of patients. They found at multivariable analysis that SOFA score and CPR were independent predictor of 1-year mortality; again, pre-Htx ECLS was not an independent determinant of that.

The Authors should be congratulated for the impressive results. However, I think that the study suffers significant limitations and lacks scientific accuracy, therefore this countertrend message is not substantiate by the present analysis, because:

  1. This is a retrospective analysis over a long period (15 years), when also Institutional guidelines for ECLS changed, according to the Author’s claim that Heart Team approach was established only in 2014 (lines 75 -77)
  2. There is no power analysis, therefore it is highly probable that a minimum 80% power is not reached for statistical analysis
  3. The 2 populations (ECLS vs no-ECLS) are significantly different in several baseline conditions, therefore they could not be compared without addressing it. I would expect a propensity-score matched population before any comparison….
  4. There is no risk-adjusted Cox-analysis but only a Kaplan-Meier comparison of two significantly different populations
  5. Indications to ECLS are not well stated and clarified: ECLS was implanted either in chronic and acute heart failure, which represent 2 completely different scenarios. A further element for an excessive case-mix and potential bias…
  6. Left ventricular venting was always achieved by atrial septostomy in only 17 patients. Furthermore, no direct LV venting was ever employed (apical venting, Impella, etc.), thus wondering that severe refractory shock really needing ECLS as a bridge was possibly reported in only a limited part of the population.

Round 2

Reviewer 1 Report

I appreciate the response of the authors to all my comments and I feel that they made significant improvement to the manuscript. My only minor comment is that it reamins difficult, at least in Western countries, to bridge directly a patients from ECMO to HTx, for several reasons. Indeed, this type of patients are really the "bottom of the barrell" and perioperative mortality and morbidity are the worse in the population of HTx patients receiving MCS. However, it is important to share data on this group of patients, supporting the concept that in very selective situation, this option can be pursued with acceptable results.

Reviewer 2 Report

Despite the Authors have partially improved the manuscript by IPWT, there are still major unsolvable issues, such as those related to power analysis, excessive case mix, different indications (acute vs chronic HF), employed techniques (atrial vs ventricular venting). A major risk for apples vs oranges comparison is still present. In particular my previous concerns related to issues n. 1, 2, 5, 6 of the first round of revision are still unsolved and they represent major bias.
